# The association between social factors and COVID-19 protective behaviors and depression and stress among midwestern US college students

Edlin Garcia Colato[1]*, Christina Ludema[2], Molly Rosenberg[2], Sina Kianersi[2], Maya Luetke[3], Chen Chen[2], Jonathan T. Macy[1]

**1** Department of Applied Health Science, Indiana University School of Public Health Bloomington, Bloomington, Indiana, United States of America, **2** Department of Epidemiology and Biostatistics, Indiana University School of Public Health Bloomington, Bloomington, Indiana, United States of America, **3** Institute for Social Research and Data Innovation, University of Minnesota, Minneapolis, Minnesota, United States of America

* eggarcia@iu.edu

**Data Availability Statement:** All relevant data are within the paper and its Supporting Information files.

## Abstract

### Purpose

The aim of this cross-sectional study was to examine the relationship between social factors and COVID-19 protective behaviors and two outcomes: depressive and perceived stress symptoms.

### Methods

In September 2020, 1,064 randomly selected undergraduate students from a large midwestern university completed an online survey and provided information on demographics, social activities, COVID-19 protective behaviors (i.e., avoiding social events and staying home from work and school), and mental health symptoms. Mental health symptoms were measured using the Center for Epidemiological Studies Depression-10 questionnaire for depression and the Perceived Stress Scale-10 for stress symptoms.

### Results

The results showed respondents who were males and also the respondents who were "hanging out" with more people while drinking alcohol reported significantly lower depressive symptoms and lower stress symptoms. On the contrary, staying home from work or school "very often" was associated with higher stress symptoms, compared with "never/rarely" staying home from work/school. Similarly, having a job with in-person interaction was also associated with increased stress.

### Conclusions

These findings suggest that lack of social engagement was associated with depression and stress symptoms among college students during the COVID-19 pandemic. Planning social

**Funding:** The study was funded by a private donation to the Indiana University Foundation. The funding was received by MR, CL, and JM The funders had no role in study design, data collection and analysis, decision to publish, or preparation of the manuscript.

**Competing interests:** The authors have declared that no competing interests exist.

activities that align with recommended safety precautions, as well as meet students' social needs, should be an important priority for higher education institutions.

## Introduction

### Background

With the emergence of the Coronavirus Disease 2019 (COVID-19) pandemic, the public received stay-at-home orders and recommendations to wear masks, reduce participation in large social events, and maintain six feet of physical distance, among other protective behaviors to reduce the risk of severe acute respiratory syndrome coronavirus 2 (SARS-CoV-2) transmission. In addition to the risk to physical health posed by the virus, approximately 80% of adults reported that the COVID-19 pandemic was a significant source of stress [1]. Young adults in college are a particularly vulnerable group. During college, young adults may experience a variety of mental health symptoms, such as those related to depression and stress, as well as experiment with substances such as e-cigarettes/vaping and alcohol [2]. Mild to severe symptoms of depression are highest among 18 to 29-year-olds (21%) and females (21.8%) [3]. Stressors for college students may involve academic pressures, financial worries, and family obligations, with the COVID-19 pandemic adding to that stress. In a study of Australian university students conducted before the COVID-19 pandemic, field of study, hours of study per week, hours of paid work per week, and number of hours caring for family per week were found to be associated with increased severity of depression and stress [4]. Furthermore, many undergraduates are employed [5, 6] to help afford courses, housing, or other living expenses, which could also contribute to stress. Acute stressors are a part of everyday life, but chronic stress can lead to more serious conditions such as anxiety or depression [7]. Significant increases in perceived stress and depression were found among undergraduate students at a large Northeastern US university in Spring 2020 during the start of the pandemic [8].

College student life is also characterized by the consumption of alcohol. Drinking alcohol is viewed as a social activity, influenced by time spent with peers [9]. Alcohol is the most frequently used substance, reportedly consumed by 53.5% of full-time US college students ages 18–22 [10]. During the COVID-19 lockdown, Belgian college students were found to have significantly reduced their alcohol use [11]. Similarly, e-cigarette use, which has also been on the rise among young adults, saw a decrease in use during the beginning of the COVID-19 pandemic, yet some young adults changed their overall substance use habits [12, 13]. These reductions in substance use could be suggestive of fewer opportunities for social gatherings among students as a result of COVID-19 prevention policies [14].

The COVID-19 pandemic has brought new challenges to the efforts of keeping college students safe and healthy. There has been limited research related to past epidemics and their effects on mental health [15], making it difficult to discern the extent of the impact of the COVID-19 pandemic on college students' psychological health [16]. Studies focused on college student mental health during the earlier days of the COVID-19 pandemic found that social support was associated with better psychological health [17]. Compared to other age groups, young adults were found to be at highest risk of loneliness as a result of the pandemic [18]. The mental health of students has been identified as a top concern for college and university leadership [19]. A recent study involving college students from seven US states found that those between the ages 18 to 24, women, those with fair/poor general health, those who had eight or more daily hours of screen time, or those who knew someone infected with COVID-

19 all had higher levels of psychological impact [20]. In a study of Chinese college students, academic workload, separation from school, and fears of infection were found to have negative effects on perceived stress [21].

However, it is unknown whether social activities (e.g., hanging out with friends while drinking) and COVID-19 protective behaviors (e.g., staying home from work or school, or avoiding social events) are associated with symptoms of depression and stress among young adults in college. A particular challenge is finding the right balance between having people engage in COVID-19 protective behaviors, but without it leading to negative consequences for their mental health [16]. COVID-19 mental health studies have primarily examined the relationships between behaviors and mental health symptoms during the beginning months of the pandemic [18, 20–25]. Few have assessed US college students' mental health in Fall 2020 during the COVID-19 pandemic as it relates to social factors and COVID-19 protective behaviors [26, 27].

There is a need for this research because the pandemic is still ongoing, and we continue to be at risk for future pandemics. Although there is substantial existing literature on mental health and college students, there is a research gap around sociodemographic and behavioral activities and their relationship with students' mental health during the pandemic context. This knowledge is valuable to support university efforts focused on improving the mental health of college students.

## Objectives

The primary objective of the current study was to examine the relationship between social factors and COVID-19 protective behaviors and two measures of mental health status: depressive and stress symptoms. We used the Strengthening the Reporting of Observational Studies in Epidemiology (STROBE) for cross-sectional studies checklist.

## Materials and methods

### Study design

A cross-sectional study design was used to identify the prevalence of depressive and stress symptoms among undergraduates in September 2020 (beginning of the fall semester). All data were retrieved from the online self-reported baseline survey collected through REDCap (Research Electronic Data Capture) [28, 29]. Participants received up to $30 for completing all the longitudinal parent study activities, from which they would have received $10 for completing the baseline survey information used in this study. This investigation was performed in accordance with the ethical standards laid down in the 1964 Declaration of Helsinki and its later amendments. The study design was reviewed by an appropriate ethical committee and received Institutional Review Board approval (Protocol #2008293852) from the Indiana University Human Subjects Office. Voluntary electronic written informed consent of the participants was obtained after the nature of the procedures had been fully explained.

### Setting

Undergraduate students at a large midwestern university were randomly sampled to participate in a parent SARS-CoV-2 antibody study during the fall 2020 semester using simple random sampling [30]. Participants completed baseline surveys between September 8 and 30, 2020. This study includes only the baseline survey responses.

## Participants

The Office of the Vice Provost for Undergraduate Education generated a random list of under-graduates to be representative of the undergraduate student population (~32,620). To be eligible for the parent antibody study, participants had to be 1) enrolled as an undergraduate student, 2) aged 18 years or older, and 3) residing in the same county where the university is located [30].

## Variables

**Outcome (1): Depression (CES-D-10).** The 10-item Center for Epidemiological Studies Depression (CES-D-10) previously validated questionnaire was used to capture self-reported symptoms of depression [31–33]. Responses to the 10 questions (0 = "rarely or none of the time (less than 1 day)" to 3 = "all of the time (5–7 days)") were summed. Possible total scores ranged from 0 to 30. Based on previous literature [32, 33], the cut-off point of 10 was used for the CES-D-10 to identify clinically significant depressive symptoms, which was shown in the previous validation studies to be equivalent to the cut-off value used in the original CES-D-20 questionnaire [33]. The CES-D-20 has been validated on young adults and college students ages 18–25 [34]. Summed scores below 10 were categorized as 0 = "no significant depressive symptoms" (reference group) and the sum of scores equaling 10 or above were categorized as 1 = "significant depressive symptoms" and were entered into the logistic regression model as a binary outcome. Cronbach's alpha for the CES-D-10 scale was 0.84.

**Outcome (2): Stress (PSS-10).** Symptoms related to stress were captured using the 10-item Perceived Stress Scale (PSS-10), a reliable and previously validated instrument [35]. All ten questions had a 5-point scale, from 0 = "never" up to 4 = "very often," four of which were positively stated and therefore reverse coded before scores were summed. Scores ranging from 0 to 13 were coded as 0 = "low stress"; scores 14 to 26 were coded as 2 = "moderate stress"; and scores between 27 and 40 were coded as 3 = "high stress" for descriptive purposes [36]. Higher scores are associated with increased perceived stress. The PSS-10 scores are categorized for descriptive purposes and do not translate into clinical diagnostic significance [37]. Therefore, for the analyses, summed scores for stress symptoms were entered into a linear regression model as the outcome variable. Cronbach's alpha for the PSS-10 was 0.85.

**Covariates.** *Sex.* On the baseline survey, participants were asked, "*What sex were you assigned at birth, on your original birth certificate?*" with options "male" and "female."

*Academic factors.* Participants reported total overall credit hours and total in-person credit hours using a drop-down menu ranging from zero to 30; both were recorded as continuous variables.

*Social factors.* Participants were asked whether they had a job or internship that involved in-person interactions (1 = yes vs. 0 = no). Participants were also asked whether they "*ever used any of the following inhaled tobacco products before today? Cigarettes, e-cigarettes, other inhaled products, and none of the above*" and were asked to mark all that apply. Past 30-day e-cigarette use was categorized into 0 = "none/zero days," 1 = "1–5 day(s)," and 2 = ">5 days" from responses ranging from 1 to 30 for the "*During the last 30 days, on how many days did you use e-cigarettes?*" question. Related to alcohol drinking behaviors, participants were asked to enter a numerical value of the number (ranging from 0 to 1,000) of persons/partners they typically "hung out" with while drinking alcohol. Non-alcohol drinkers were recoded with zero for the number of drinking partners. A separate group that was restricted to only the participants that reported drinking alcohol was also created for the reported number of drinking partners/ number of people the participants "hung out" with.

*COVID-19 protective behaviors.* Participants were asked if they practiced a series of protective behaviors in the past 7 days to prevent infection of COVID-19. The two included in this

study are: how often they "*avoided a social event I wanted to attend*" followed by how often they "*stayed at home from work/school*" with options 1 = "always," 2 = "very often," 3 = "some-times," 4 = "rarely," and 5 = "never" for both questions. The responses to both questions were reverse coded, and "rarely" and "never" were collapsed.

## Data analysis

The following models were conducted: 1) unadjusted and sex-adjusted logistic regression models for the relationship between each of the four social and behavioral variables and depressive symptoms, and 2) unadjusted and sex-adjusted linear regression models to test for the association between the four social and behavioral variables and stress symptoms. Sex is associated with both prevalence and severity of reported depressive symptoms [3] and differences in stress levels [38]. Therefore, sex was controlled for in the adjusted models. The data analyses were generated using Stata software, version 16.1 (College Station, Texas). The sample size calculation for the parent RCT study was calculated for the parent study aims [29]; however, there was no sample size calculation conducted for this current study's analysis of the baseline survey data.

## Results

### Participants

Of the 7,499 undergraduate students randomly sampled to participate in the COVID-19 antibody study, 3,430 were ineligible because they were not living in the same county as the university or not enrolled as an undergraduate student. Of the 4,069 eligible individuals, 1,397 (34.4%) provided voluntary electronic consent. One hundred thirty-three discontinued their involvement in the study by not answering any of the survey questions and were therefore excluded from this analysis.

For the mental health symptoms outcomes, 108 of the 1,264 participants had missing values for the CES-D-10 and PSS-10 questions and were removed. Of the 1,156 with complete mental health symptoms data, 91 (7.9%) participants had missing self-reported background information: age (missing 68), sex at birth (missing 1), school year (missing 3), credit hours (missing 1), in person credit hours (missing 11), job with in-person interaction (missing 2), number of alcohol drinking partners (missing 3), and COVID-19 protective behaviors (missing 2) and were removed. One participant entered zero for the total credits enrolled in and was also removed, yielding n = 1,064 for the final sample.

### Descriptive data

Table 1 summarizes the socio-demographic characteristics of the participants by both depressive and stress symptoms. Most respondents were female, White, non-Hispanic/Latinx, had none/zero day(s) of past 30-day e-cigarette use, and did not have a job or internship with in-person interaction. Of the 721 students that reported drinking alcohol, 705 (97.8%) reported "hanging out" with people while drinking. Of the students with an in-person facing job or internship, the majority (219; 74.0%) were females.

### Outcome data

Four hundred forty-seven (42.0%) participants reported having significant depressive symptoms. Based on the PSS-10, most of the respondents (705; 66.3%) experienced moderate stress symptoms, followed by 23.9% with low stress and 9.9% with high stress. The mean score on the PSS-10 was 18.28 (SD 6.52) and ranged from zero to 38.

**Table 1. Depressive symptoms and perceived stress scores in a sample of undergraduate students from a midwestern US university, Fall 2020.**

| | | Depressive Symptoms | | Perceived Stress Score | | | | |
|---|---|---|---|---|---|---|---|---|
| | N | No (%) | Yes (%) | Mean (SD) | Range | Low Stress n (%) | Moderate Stress n (%) | High Stress n (%) |
| **All** | 1,064 | 617 (58.0) | 447 (42.0) | 18.28 (6.52) | 0–38 | 254 (23.9) | 705 (66.3) | 105 (9.9) |
| **Age,** mean (SD) | 20.03 (2.49) | 20.11 (2.88) | 19.91 (1.82) | | | 20.30 (3.70) | 19.96 (2.00) | 19.83 (1.68) |
| **Sex** | | | | | | | | |
| Female | 686 (64.5) | 350 (56.7) | 336 (75.2) | 19.41 (6.43) | 2–38 | 130 (51.2) | 471 (66.8) | 85 (80.9) |
| Male | 378 (35.5) | 267 (43.3) | 111 (24.8) | 16.22 (6.16) | 0–35 | 124 (48.8) | 234 (33.2) | 20 (19.0) |
| **Gender** (n = 1,063) | | | | | | | | |
| Female | 675 (63.5) | 349 (56.7) | 326 (72.9) | 19.33 (6.42) | 2–38 | 130 (51.2) | 465 (66.0) | 80 (76.2) |
| Male | 375 (35.3) | 266 (43.2) | 109 (24.4) | 16.19 (6.14) | 0–35 | 124 (48.8) | 232 (32.9) | 19 (18.1) |
| Transgender | 4 (0.4) | 1 (0.2) | 3 (0.7) | 22.50 (5.51) | 17–29 | 0 | 3 (0.4) | 1 (0.9) |
| None of the above | 9 (0.8) | 0 | 9 (2.0) | 25.67 (5.36) | 16–32 | 0 | 4 (0.6) | 5 (4.8) |
| **Race** | | | | | | | | |
| African American/Black | 17 (1.6) | 11 (1.8) | 6 (1.3) | 16.00 (8.28) | 5–30 | 9 (3.5) | 6 (0.8) | 2 (1.9) |
| Asian | 77 (7.2) | 47 (7.6) | 30 (6.7) | 17.99 (6.20) | 5–35 | 18 (7.1) | 54 (7.7) | 5 (4.8) |
| Multiracial | 53 (5.0) | 33 (5.3) | 20 (4.5) | 17.40 (7.03) | 0–35 | 15 (5.9) | 33 (4.7) | 5 (4.8) |
| Other | 50 (4.7) | 23 (3.7) | 27 (6.0) | 19.22 (5.57) | 3–29 | 6 (2.4) | 39 (5.5) | 5 (4.8) |
| White | 867 (81.5) | 503 (81.5) | 364 (81.4) | 18.35 (6.52) | 0–38 | 206 (81.1) | 573 (81.3) | 88 (83.8) |
| **Ethnicity** | | | | | | | | |
| Hispanic/Latinx | 85 (8.0) | 47 (7.6) | 38 (8.5) | 18.88 (5.47) | 6–35 | 10 (3.9) | 68 (9.6) | 7 (6.7) |
| Non-Hispanic/Latinx | 979 (92.0) | 570 (92.4) | 409 (91.5) | 18.23 (6.59) | 0–38 | 244 (96.1) | 637 (90.4) | 98 (93.3) |
| **School Year** | | | | | | | | |
| First | 239 (22.5) | 146 (23.7) | 93 (20.8) | 17.48 (6.67) | 0–37 | 67 (26.4) | 151 (21.4) | 21 (20.0) |
| Second | 229 (21.5) | 119 (19.3) | 110 (24.6) | 19.37 (6.52) | 3–35 | 45 (17.7) | 154 (21.8) | 30 (28.6) |
| Third | 260 (24.4) | 151 (24.5) | 109 (24.4) | 18.53 (6.51) | 0–35 | 59 (23.2) | 176 (25.0) | 25 (23.8) |
| Fourth | 309 (29.0) | 187 (30.3) | 122 (27.3) | 17.87 (6.39) | 1–38 | 79 (31.1) | 204 (28.9) | 26 (24.8) |
| Fifth + | 27 (2.5) | 14 (2.3) | 13 (2.9) | 18.37 (5.54) | 11–31 | 4 (1.6) | 20 (2.8) | 3 (2.9) |
| **Total Credit Hours** mean (SD) | 15.19 (2.17) | 15.17 (2.08) | 15.22 (2.29) | | | 15.20 (2.17) | 15.17 (2.17) | 15.28 (2.22) |
| **In-person Credit hours** mean (SD) | 2.99 (2.97) | 3.06 (3.01) | 2.88 (2.91) | | | 3.00 (2.90) | 3.02 (3.02) | 2.74 (2.82) |
| **Job or Internship†** | | | | | | | | |

*(Continued)*

**Table 1.** (Continued)

| | N | Depressive Symptoms | | Perceived Stress Score | | | | |
| | | No (%) | Yes (%) | Mean (SD) | Range | Low Stress n (%) | Moderate Stress n (%) | High Stress n (%) |
|---|---|---|---|---|---|---|---|---|
| No | 768 (72.2) | 464 (75.2) | 304 (68.0) | 17.72 (6.50) | 0–38 | 201 (79.1) | 50 (71.1) | 66 (62.9) |
| Yes | 296 (27.8) | 153 (24.8) | 143 (32.0) | 19.75 (6.34) | 2–35 | 53 (20.9) | 204 (28.9) | 39 (37.1) |
| **Past 30-day E-cigarette Use** | | | | | | | | |
| None/zero day(s) | 762 (71.6) | 450 (72.9) | 312 (69.8) | 18.15 (6.62) | 0–38 | 187 (73.6) | 500 (70.9) | 75 (71.4) |
| 1–5 days | 101 (9.5) | 56 (9.1) | 45 (10.1) | 18.37 (6.00) | 6–30 | 21 (8.3) | 70 (9.9) | 10 (9.5) |
| > 5 days | 201 (18.9) | 111 (18.0) | 90 (20.1) | 18.75 (6.36) | 3–37 | 46 (18.1) | 135 (19.1) | 20 (19.0) |
| **Number of People Hanging out with While Drinking Alcohol**, mean (SD) | 3.82 (4.78) | 4.44 (5.41) | 2.97 (3.58) | | | 4.12 (4.64) | 3.90 (5.00) | 2.54 (3.19) |
| **Avoided Social Events Interested in Attending‡** | | | | | | | | |
| Always | 316 (29.7) | 169 (27.4) | 147 (32.9) | 19.02 (6.72) | 0–37 | 68 (26.8) | 206 (29.2) | 42 (40.0) |
| Very Often | 283 (26.6) | 156 (25.3) | 127 (28.4) | 18.30 (6.32) | 3–35 | 68 (26.8) | 191 (27.1) | 24 (22.9) |
| Sometimes | 275 (25.8) | 168 (27.2) | 107 (23.9) | 18.01 (6.43) | 0–38 | 66 (26.0) | 187 (26.5) | 22 (20.9) |
| Never/Rarely | 190 (17.9) | 124 (20.1) | 66 (14.8) | 17.41 (6.49) | 2–35 | 52 (20.5) | 121 (17.2) | 17 (16.2) |
| **Stayed Home from Work/School‡** | | | | | | | | |
| Always | 266 (25.0) | 157 (25.4) | 109 (24.4) | 18.37 (6.92) | 0–38 | 62 (24.4) | 175 (24.8) | 29 (27.6) |
| Very Often | 198 (18.6) | 105 (17.0) | 93 (20.8) | 19.29 (6.31) | 0–35 | 36 (14.2) | 137 (19.4) | 25 (23.8) |
| Sometimes | 170 (16.0) | 97 (15.7) | 73 (16.3) | 18.13 (6.16) | 4–35 | 38 (15.0) | 117 (16.6) | 15 (14.3) |
| Never/Rarely | 430 (40.4) | 258 (41.8) | 172 (38.5) | 17.82 (6.46) | 2–35 | 118 (46.5) | 276 (39.1) | 36 (34.3) |

*Note*.

†with in-person interaction

‡Measure taken in the last 7 days to prevent infection from COVID-19; CES-D-10 <10 = no; CES-D-10 >10 = yes; PSS-10 0–13 = low stress; PSS-10 14–26 = moderate stress; PSS-10 27–40 = high stress

## Main results

**Social/Behavioral predictors of depressive symptoms in the unadjusted analyses.** As shown in Table 2, males were less likely than females to report significant depressive symptoms. Similarly, students with an in-person facing job or internship were more likely to report significant depressive symptoms than participants without an in-person facing job or internship. However, the number of total credit hours and in-person credit hours were not associated with increased depressive symptoms.

Past 30-day e-cigarette use was also not found to be associated with depressive symptoms. In contrast, the number of people the students "hungout" with while drinking alcohol was negatively associated with depressive symptoms. Similarly, among the restricted sample of

**Table 2. Results for logistic regression for depressive symptoms (CES-D-10) and linear regression for stress symptoms (PSS) by predictors (95% CI), Undergraduate Students from a midwestern US university, Fall 2020.**

| | Depressive Symptoms | | Stress Symptoms | |
|---|---|---|---|---|
| | Unadjusted OR | Adjusted[a] OR | Unadjusted $\beta$ | Adjusted[a] $\beta$ |
| **Sex** | | | | |
| Female | Ref | - - | Ref | - - |
| Male | **0.43 (0.33, 0.56)** *** | - - | **-3.19 (-3.99, -2.40)**\*** | - - |
| **Total Credit Hours** | 1.01 (0.96, 1.07) ns | 1.01 (0.95, 1.07) ns | 0.002 (-0.18, 0.18) ns | -0.02 (-0.19, 0.16) ns |
| In-person Credit Hours | 0.98 (0.94, 1.02) ns | 0.98 (0.94, 1.02) ns | -0.07 (-0.20, 0.06) ns | -0.07 (-0.19, 0.06) ns |
| **Job or Internship†** | | | | |
| No | Ref | Ref | Ref | Ref |
| Yes | **1.43 (1.09, 1.87)*** | 1.29 (0.98,1.71) ns | **2.03 (1.16, 2.90)** ** | **1.63 (0.78, 2.49)*** |
| **Past 30-day E-cigarette Use** | | | | |
| None/0 days | Ref | Ref | Ref | Ref |
| 1–5 days | 1.16 (0.76, 1.76) ns | 1.22 (0.80, 1.88) ns | 0.22 (-1.13, 1.57) ns | 0.40 (-0.91, 1.72) ns |
| > 5 days | 1.17 (0.85, 1.60) ns | 1.29 (0.94, 1.78) ns | 0.60 (-0.41, 1.61) ns | 0.95 (-0.03, 1.94) ns |
| **Number of People Hanging out with While Drinking Alcohol** | **0.92 (0.89, 0.95)*** | **0.92 (0.89, 0.95)*** | **-0.13 (-0.22, -0.06)** ** | **-0.13 (-0.21, -0.05)** ** |
| **Avoided Social Events Interested in Attending** | | | | |
| Never/Rarely | Ref | Ref | Ref | |
| Sometimes | 1.20 (0.81, 1.76) ns | 1.09 (0.74, 1.62) ns | 0.60 (-0.60, 1.81) ns | 0.23 (-0.95, 1.40) ns |
| Very often | **1.53 (1.05, 2.23)*** | 1.38 (0.94, 2.03) ns | 0.89 (-0.31, 2.09) ns | 0.44 (-0.73, 1.61) ns |
| Always | **1.63 (1.13, 2.37)*** | 1.40 (0.96, 2.05) ns | **1.61 (0.44, 2.78)** ** | 0.95 (-0.21, 2.10) ns |
| **Stayed Home from Work/School** | | | | |
| Never/Rarely | Ref | Ref | Ref | Ref |
| Sometimes | 1.13 (0.79, 1.62) ns | 1.12 (0.78, 1.61) ns | 0.31 (-0.85, 1.46) ns | 0.26 (-0.86, 1.38) ns |
| Very often | 1.33 (0.95, 1.86) ns | 1.33 (0.94, 1.89) ns | **1.46 (0.37 2.56)** ** | **1.44 (0.37, 2.51)** ** |
| Always | 1.04 (0.76, 1.42) ns | 1.05 (0.77, 1.45) ns | 0.54 (-0.45, 1.54) ns | 0.58 (-0.38, 1.55) ns |

[a]Adjusted for sex

†with in-person interaction; *ns* = not significant

*p-value <0.05

**<0.01

***<0.001

students who reported drinking, the number of people the students "hung out" with while drinking alcohol was also negatively associated with depressive symptoms. On the other hand, those who avoided social events either "very often" or "always" had higher odds of reported significant depressive symptoms compared to those who reported "never/rarely" avoiding those social events.

**Social/Behavioral predictors of depressive symptoms in the sex-adjusted analyses.** No significant relationship was found between the number of total credit hours and in-person credit hours and depressive symptoms, after adjusting for sex. However, sex largely accounted for the association between having an in-person facing job or internship and significant depressive symptoms. After adjusting for sex, there was no longer a significant association between having an in-person facing job or internship and depressive symptoms.

The relationship between past 30-day e-cigarette use and depressive symptoms remained relatively unchanged and was still not significant, after controlling for sex. The significant association between the number of people the students "hung out" with while drinking alcohol and depressive symptoms remained unchanged after controlling for sex in the model.

Likewise, in the restricted sample of participants who reported drinking alcohol, the significant association between the number of people the students "hung out" with while drinking alcohol and depressive symptoms was also identical after controlling for sex. However, both of the COVID-19 protective behaviors (avoiding social events and staying home from work and school) were not significantly associated with having significant depressive symptoms when controlling for sex in the models.

**Social/Behavioral predictors of stress symptoms in the unadjusted analyses.**    As shown in Table 2, males were significantly less likely to report stress symptoms compared to females. Number of total credit hours and in-person credit hours were not associated with increased stress symptoms. However, having a job or internship with in-person interaction was associated with reporting greater stress symptoms.

There was no statistically significant relationship between the past 30-day e-cigarette use and stress. Students who reported hanging out with an increased number of people while drinking reported less stress; the same was true for the restricted sample of students who reported drinking. Among the restricted sample of students who reported drinking, the increased number of reported people "hanging out" with while drinking alcohol was similarly negatively associated with stress.

Among the COVID-19 protective behaviors, participants who "always" avoided social events reported greater stress symptoms compared to those who "never/rarely" attended social events. In addition, staying home from work/school "very often" was associated with reporting greater stress symptoms compared to "never/rarely" staying home from work/school.

**Social/Behavioral predictors of stress symptoms in the sex-adjusted analyses.**    After controlling for sex, there was no relationship between the number of total credit hours and in-person credit hours and stress symptoms. However, the association between having a job or internship with in-person interaction and increased stress remained significant. The relationship between the increased number of people the respondents "hung out" with while drinking alcohol and increased stress symptoms also remained significant. Similarly, among the restricted sample of participants who reported drinking alcohol, the increased number of people the respondents "hung out" with while drinking alcohol and increased stress was significant after controlling for sex. Lastly, the relationship between "always" staying home from work/ school compared to "never/rarely" remained significant after controlling for sex in the model.

## Discussion

In this study, we found that fewer than half of the undergraduate student participants reported depressive symptoms, and the majority reported moderate to high levels of stress symptoms in Fall 2020. We also found that key social and behavioral variables were associated with these negative health outcomes. Namely, these included having a job or internship with in-person interaction, the number of people students "hung out" with while drinking alcohol, avoiding social events, and staying home from work or school. The relationship between the number of people students 'hung out' with while drinking alcohol and both depressive and stress symptoms persisted even after accounting for the expected and strong sex differences in our two mental health outcomes.

The percentage of reported high stress scores in our sample (9.9%) was lower than the July/ August 2020 PSS-10 scores from undergraduates at a Southeastern US university where 26.9% reported high stress [25]. Besides the difference in geographical region, the difference in high stress scores reported by the two populations could be attributed to the reduced social constraints and return back to campus that followed the summer of 2020. Interestingly, academic workload in the form of total credit hours and in-person credit hours were not found to have a

statistically significant association with either depressive or stress symptoms. This could be due to the added university safety features requiring students to wear masks and social distance, as well as the enhanced cleaning procedures and limited classroom capacity for in-person lectures, which may have helped prevent any significant stress. It should also be noted that undergraduates were considered full-time with an enrollment of at least 12 credit hours per semester. However, the University recommends that undergraduates complete approximately 15 to 16 credit hours during the fall and spring semesters to complete their degree in four years. Therefore, on average, students were enrolled in the recommended amount of 15 credit hours.

For some students, returning to the university came at a financial cost that required taking on either an internship or part-time job to survive financially. Participants who reported having a job or internship with in-person interaction reported significantly increased stress. Students employed or interning in a position that required interacting with people in-person were potentially presented with unique stressors brought on by balancing work responsibilities, as they also had to engage in COVID-19 protective behaviors. This finding is significant because as universities make efforts to protect their students, there are limitations to those efforts such as the workplace that falls outside of their jurisdiction. Possible recommendations include outreach programs dedicated to providing behavioral health, financial, and academic support to the students who are employed in client-facing jobs.

For the COVID-19 protective behaviors, there was significant increased perceived stress among participants who reported they "very often" stayed home from work/school compared to those who "rarely/never" stayed home from work/school. However, this association was not observed for participants who reported "always" staying home. The undergraduates who reported staying home "very often" may have had ambivalent feelings towards whether to stay home that might have created stress. Alternatively, the fear of contagion as a result of those few occasions on which they went out might have also led to increased stress. Future studies might consider exploring this further by collecting qualitative data on individuals' feelings and perceptions regarding the risk and stress of social engagement activities during the pandemic.

Aspects of our study design influence the interpretation of our study findings. First, as an observational cross-sectional study, there are issues with directionality of effects. Responses were collected at a single point in time, near the start of the semester, making it difficult to discern whether other unreported factors were associated with the selected social and behavioral activities or their mental health symptoms. Furthermore, we cannot determine whether increased social engagement led to reduced depressive symptoms or whether decreased social engagement was a consequence of those mental health symptoms. Aside from consuming alcohol, we are unaware of what additional in-person social exchanges occurred. Previous research found that very frequent in-person social connections during the pandemic were associated with lower depression [39].

Another limitation of this study was the use of self-reported data collected via an online survey instrument. There is a possibility of bias due to under-reporting for some of our key selected variables. For instance, due to the legal age of tobacco and nicotine products having been raised to 21, matching the legal age of consumption for alcohol, there may have been some under-reporting of substance use of individuals between age 18 and 20 [40]. However, self-response surveys for nicotine use and mental health symptoms are conventional for this type of research [41, 42]. Further, an additional limitation of our cross-sectional design of the study is the inability to capture the fluidity of mental health symptoms. Lastly, unlike in the Charles et al. study [26], there were no pre-pandemic matched mental health scores to compare changes in symptoms among our sample.

During the regular college experience, college students experience a breadth of challenges, whether personal, academic, financial, or otherwise. The COVID-19 pandemic worsened

some of those existing challenges and introduced new ones. Having conducted this study at the start of the Fall 2020 academic semester, as students returned from an all-online curriculum to a hybrid learning mode, is not representative of all the different stages of the pandemic. However, it offers some insight into students' mental health and well-being for future public health crises.

Strengths of this study includes the simple random sampling of the students, reducing the risk of sampling bias. However, it was vulnerable to volunteer bias. It is possible that eligible students who were experiencing extreme hardships were unable to participate in this study. Therefore, although our study findings might be representative of the general undergraduate student body, it might not be generalizable to some subgroups that were at severe risk of depression or stress.

The findings of this current study may have some implications for future university public health communications and prevention efforts. Universities need to provide opportunities for social interaction that maintain safety from infectious disease transmission. For example, this may include hosting social activities in an outdoor space where students can maintain a safe distance from each other while interacting with one another, rather than staying in their dorm or apartment in social isolation. This may also include establishing peer support groups that regularly meet and provide each other with guidance on how to safely socialize. Our study contributes to the existing mental health and COVID-19 among college student literature by identifying a negative relationship between the recommended refrain of social activities and protective behaviors and two mental health outcomes: depression and stress during a time when the requirements were no longer as strongly enforced.

Future research could further investigate the relationship between these social factors, COVID-19 protective behaviors, and the mental health of students who, now years into the pandemic, have continued to refrain from social activities. Furthermore, additional research could review university pandemic incident response plans and the available support systems and existing interventions aimed at preventing depression and stress.

## Conclusion

These findings suggest that social engagement was associated with lower levels of depressive symptoms among college students during the fall 2020 semester. It may be that social engagement acted as a protective factor for depression during this time. Offering social activities that align with recommended safety precautions and meet students' social needs should be considered as an important priority for higher education institutions as they continue to address the COVID-19 pandemic and plan for future public health emergencies. This study provides additional information about undergraduate mental health outcomes during the ongoing pandemic to inform policies, interventions, and the provision of services needed to address top college and university leadership concerns.

## Supporting information

**S1 Data.**
(XLSX)

## Author Contributions

**Conceptualization:** Edlin Garcia Colato, Christina Ludema, Molly Rosenberg, Jonathan T. Macy.

**Data curation:** Edlin Garcia Colato.

**Formal analysis:** Edlin Garcia Colato, Christina Ludema, Molly Rosenberg, Jonathan T. Macy.

**Funding acquisition:** Christina Ludema, Molly Rosenberg, Jonathan T. Macy.

**Investigation:** Edlin Garcia Colato, Christina Ludema, Molly Rosenberg, Sina Kianersi, Maya Luetke, Chen Chen, Jonathan T. Macy.

**Methodology:** Edlin Garcia Colato, Christina Ludema, Molly Rosenberg, Jonathan T. Macy.

**Project administration:** Christina Ludema, Molly Rosenberg, Jonathan T. Macy.

**Resources:** Christina Ludema, Molly Rosenberg, Jonathan T. Macy.

**Software:** Edlin Garcia Colato.

**Supervision:** Christina Ludema, Molly Rosenberg, Jonathan T. Macy.

**Validation:** Edlin Garcia Colato.

**Visualization:** Edlin Garcia Colato, Jonathan T. Macy.

**Writing – original draft:** Edlin Garcia Colato.

**Writing – review & editing:** Edlin Garcia Colato, Christina Ludema, Molly Rosenberg, Sina Kianersi, Maya Luetke, Chen Chen, Jonathan T. Macy.

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
