## [Decision Letter · Decision Letter 0]

14 Sep 2022

PONE-D-22-20107Social factors associated with college students' depression and stress during the COVID-19 pandemic: A cross-sectional studyPLOS ONE

Dear Dr. Garcia Colato,

Thank you for submitting your manuscript to PLOS ONE. After careful consideration, we feel that it has merit but does not fully meet PLOS ONE’s publication criteria as it currently stands. Therefore, we invite you to submit a revised version of the manuscript that addresses the points raised during the review process.

We look forward to receiving your revised manuscript.

Kind regards,

Michio Murakami

Academic Editor

PLOS ONE

Journal Requirements:

2. Please state the full name of the Institutional Review Board that approved your study.

3. You indicated that you had ethical approval for your study. In your Methods section, please ensure you have also stated whether you obtained consent from parents or guardians of the minors included in the study or whether the research ethics committee or IRB specifically waived the need for their consent.

Reviewers' comments:

Reviewer's Responses to Questions

**Comments to the Author**

1. Is the manuscript technically sound, and do the data support the conclusions?

Reviewer #1: Partly

Reviewer #2: Yes

2. Has the statistical analysis been performed appropriately and rigorously? 

Reviewer #1: No

Reviewer #2: Yes

3. Have the authors made all data underlying the findings in their manuscript fully available?

Reviewer #1: No

Reviewer #2: Yes

4. Is the manuscript presented in an intelligible fashion and written in standard English?

Reviewer #1: No

Reviewer #2: Yes

5. Review Comments to the Author

Reviewer #1: General comments

The study is good, but the write-up needs restructuring and editorial improvement to be coherent and easily flow.

Major issues

The title should be self-explanatory and need to specify the place of the study. The title is not in line with the stated objective i.e. at least the relationship between social factors and COVID-19 protective behaviours is missing from the title.

The abstract did not follow the journal guideline showing introduction, method, result and conclusion. In addition, there is not any effect size reported in the abstract. As it stands now, it is wordy and includes unnecessary details.

The background section is not focused, and it does not show what were known, what were unknown and the need for the study, justification was shallow and not well developed.

The authors have mentioned that the tools used were validated. Where were the tools validated? The cited references showed that the validation is somewhere else. Tools should be validated in the country where they are used for data collection. Otherwise there might be still cross-cultural differences and the use of non-validated tool should be mentioned as limitation.

Where is the sample size calculation. as it stands now it seems that authors have approached students who were planned for COVID-19 antibody test were asked to participate in the survey. This may have its own impact in the generalizability. Detail information on how they were initially recruited for that study must be at least cited.

The study seems outdated as the data is collected in 2020. While the COVID-19 pandemic is the very pressing issue until now, there are several evidence on mental health impacts among different segments of population including students. The importance of this study should be clearly justified.

Authors have dichotomized the depressive symptoms outcome but used as it is for stress symptoms. What was the reason to go for dichotomizing in the analysis of factors for depressive symptoms?

Authors mentioned that they have used random sampling as a strength, which specific type of random sampling did they use?

What were the implications? The discussion seems straight jacketed, and it seems the repetition of the result section as it stands now.

The discussion lacks recommendation based on the results.

Authors have mentioned that they have paid up to $30 per questionnaire, introducing financial issues in survey participation has its own problem. Please comment on this.

Minor issues

Keywords should be written in alphabetical order. The use of mental health as keyword do not add any information as the mental health symptoms dealt with were only depressive symptoms and perceived stress.

Background lines 79-80, needs citation

Objectives lines 100-103, is not objective it should be moved to introduction section. In addition there are several studies on mental health problems among students all over the globe.

Lines 216 to 217 is incomplete sentence.

Line 265 “key results” is unnecessary sub heading, check journal guideline.

Line 275 to 276 repeating the objective once again in the discussion section is meaningless.

Several limitations have been mixed up with the strength of the study section in the discussion.

Tables should be named appropriately i.e. play and year has to be mentioned.

Reviewer #2: This paper is very interesting and easy to read. However, I recommend to improve the discussion and conclusion sections. The discussion should be in-depth and contains not only summaries of research results, but above all their significance and implications for university practice

6. PLOS authors have the option to publish the peer review history of their article (what does this mean?). If published, this will include your full peer review and any attached files.

Reviewer #1: No

Reviewer #2: No

---

## [Author Response · Author response to Decision Letter 0]

3 Nov 2022

Response to the Review Team

Reviewer Comment Response

1. The title should be self-explanatory and need to specify the place of the study. The title is not in line with the stated objective i.e. at least the relationship between social factors and COVID-19 protective behaviours is missing from the title. 

Response: We appreciate the reviewer’s feedback on how to improve the original title. Based on the provided feedback we have revised the original title “Social factors associated with college students' depression and stress during the COVID-19 pandemic: A cross-sectional study” to the title listed below.

New title: The association between social factors and COVID-19 protective behaviors and depression and stress among midwestern US college students

Page 1, lines 4-5 (page numbers and lines here and throughout are from the clean copy, not the tracked changes)

2. The abstract did not follow the journal guideline showing introduction, method, result and conclusion. In addition, there is not any effect size reported in the abstract. As it stands now, it is wordy and includes unnecessary details. 

Response: We thank the reviewer for the comment regarding the structure for the abstract. Upon reviewing the PLOS ONE journal submission guidelines as well as the most recently published articles, our take on PLOS ONE’s guideline for the abstract format is that the abstract is unstructured. However, as per your request we have revised the abstract to a structured format showing the purpose, methods, results, and conclusions as follows.

Purpose: The aim of this cross-sectional study was to examine the relationship between social factors and COVID-19 protective behaviors and two outcomes: depressive and perceived stress symptoms. 

Methods: In September 2020, 1,064 randomly selected undergraduate students from a large midwestern university completed an online survey and provided information on demographics, social activities, COVID-19 protective behaviors (i.e., avoiding social events and staying home from work and school), and mental health symptoms. Mental health symptoms were measured using the Center for Epidemiological Studies Depression-10 questionnaire for depression and the Perceived Stress Scale-10 for stress symptoms. 

Results: The results showed respondents who were males and also the respondents who were “hanging out” with more people while drinking alcohol reported significantly lower depressive symptoms and lower stress symptoms. On the contrary, staying home from work or school “very often” was associated with higher stress symptoms, compared with “never/rarely” staying home from work/school. Similarly, having a job with in-person interaction was also associated with increased stress. 

Conclusions: These findings suggest that lack of social engagement was associated with depression and stress symptoms among college students during the COVID-19 pandemic. Planning social activities that align with recommended safety precautions, as well as meet students’ social needs, should be an important priority for higher education institutions.

Page 2, lines 28-46 

3. The background section is not focused, and it does not show what were known, what were unknown and the need for the study, justification was shallow and not well developed. 

Response: We have re-organized the introduction to show what is known from pages 3-4 lines 52 to 94, what is unknown on page 5 lines 95 to 104, and the need for the study from lines 105 to 110. 

4. The authors have mentioned that the tools used were validated. Where were the tools validated? The cited references showed that the validation is somewhere else. Tools should be validated in the country where they are used for data collection. Otherwise there might be still cross-cultural differences and the use of non-validated tool should be mentioned as limitation. 

Response: Thank you for the comment regarding the location of the population for the validation studies. The references have been updated accordingly to show that the original CES-D 20 has been previously validated with young adults and college students who are age 18-25 located within the United States. 

“The CES-D-20 has been validated on young adults and college students ages 18-25 [34].”

Page 7, lines 148-149

As for the PSS-10, reference #35 is the validation study for the scale using US college students that showed it was a valid and reliable measure of perceived stress.

Page 7, line 155

Reference #35: Cohen S, Kamarck T, Mermelstein R. A Global Measure of Perceived Stress. J Health Soc Behav. 1983;24(4):385-96.

5. Where is the sample size calculation. as it stands now it seems that authors have approached students who were planned for COVID-19 antibody test were asked to participate in the survey. This may have its own impact in the generalizability. Detail information on how they were initially recruited for that study must be at least cited. 

Response: We thank the reviewer for their question regarding the sample size calculation. We have updated the text found in the methods-data analysis section clarifying that the sample size calculation for the parent RCT study was calculated based on the parent study aims; however, there was no sample size calculation conducted for this current study’s analysis of the baseline survey data.

Text now reads: “The sample size calculation for the parent RCT study was calculated for the parent study aims [29]; however, there was no sample size calculation conducted for this current study’s analysis of the baseline survey data.”

Page 9, lines 196 - 199

6. The study seems outdated as the data is collected in 2020. While the COVID-19 pandemic is the very pressing issue until now, there are several evidence on mental health impacts among different segments of population including students. The importance of this study should be clearly justified. 

Response: We appreciate the reviewer’s comment regarding the age of the data. Yes, we acknowledge that the data is from two years ago. However, as the pandemic is still ongoing and there is always a risk for a future pandemic, the insights learned from the time of this study are still relevant for our current situation as well as important for future events. Although there are many studies that have focused on mental health problems among students, few have focused on the relationship between the social factors and COVID-19 protective behaviors (staying at home from work/school and avoiding social events) and the mental health of university students. We have revised the sentences in the introduction and discussion to delineate the contribution of our study to existing literature.

Revised statement in the introduction: “There is a need for this research because the pandemic is still ongoing, and we continue to be at risk for future pandemics. Although there is substantial existing literature on mental health and college students, there is a research gap around sociodemographic and behavioral activities and their relationship with students’ mental health during the pandemic context. This knowledge is valuable to support university efforts focused on improving the mental health of college students.”

Page 5, lines 105-110

Revised statement in the discussion: “Our study contributes to the existing mental health and COVID-19 among college student literature by identifying a negative relationship between the recommended refrain of social activities and protective behaviors and two mental health outcomes: depression and stress during a time when the requirements were no longer as strongly enforced.” 

Page 20-21, lines 366-370

7. Authors have dichotomized the depressive symptoms outcome but used as it is for stress symptoms. What was the reason to go for dichotomizing in the analysis of factors for depressive symptoms? 

Response: Thanks for highlighting this question. In the text, we added these clarifying sentences:

The new text for CES-D-10 reads: “Based on previous literature [32, 33], the cut-off point of 10 was used for the CES-D-10 to identify clinically significant depressive symptoms…”

Page 7, lines 145-147

The new text for PSS-10 “The PSS-10 scores are categorized for descriptive purposes and do not translate into clinical diagnostic significance [37]”

Page 8, lines 161-162

8. Authors mentioned that they have used random sampling as a strength, which specific type of random sampling did they use? 

Response: We thank the reviewer for their inquiry on the specific type of random sampling used by the office that provided the random sample of potential students. Based on your feedback we have updated the text to state it was simple random sampling. 

“Undergraduate students at a large midwestern university were randomly sampled to participate in a parent SARS-CoV-2 antibody study during the fall 2020 semester using simple random sampling [30].”

Page 6, line 130-133

9. What were the implications? The discussion seems straight jacketed, and it seems the repetition of the result section as it stands now.

Response: Based on the reviewer’s feedback, we updated the discussion section by removing repetitive text from the results section and further elaborated on the significance and explanation of the findings within the discussion section. 

“This finding is significant because as universities make efforts to protect their students, there are limitations to those efforts such as the workplace that falls outside of their jurisdiction. Possible recommendations include outreach programs dedicated to providing behavioral health, financial, and academic support to the students who are employed in client-facing jobs.”

Page 18, lines 313-317

And

“The findings of this current study may have some implications for future university public health communications and prevention efforts. Universities need to provide opportunities for social interaction that maintain safety from infectious disease transmission. For example, this may include hosting social activities in an outdoor space where students can maintain a safe distance from each other while interacting with one another, rather than staying in their dorm or apartment in social isolation. This may also include establishing peer support groups that regularly meet and provide each other with guidance on how to safely socialize. Our study contributes to the existing mental health and COVID-19 among college student literature by identifying a negative relationship between the recommended refrain of social activities and protective behaviors and two mental health outcomes: depression and stress during a time when the requirements were no longer as strongly enforced.” 

Page 20, lines 360-370

10. The discussion lacks recommendation based on the results. 

Response: Thank you to the reviewer for their comment. Based on the comment, we have added a paragraph on recommendations for future research right before the conclusion section.

“Future research could further investigate the relationship between these social factors, COVID-19 protective behaviors, and the mental health of students who, now years into the pandemic, have continued to refrain from social activities. Furthermore, additional research could review university pandemic incident response plans and the available support systems and existing interventions aimed at preventing depression and stress.”

Page 21, lines 371-375

11. Authors have mentioned that they have paid up to $30 per questionnaire, introducing financial issues in survey participation has its own problem. Please comment on this. 

Response: We agree with the reviewer that it is important to be transparent about participant compensation and thoughtful about any undue influence this compensation might introduce. 

The total possible participant incentive for the longitudinal parent study was $30, with partial payments for completing partial study procedures. This compensation amount and structure was developed to reflect the time and energy involved in responding to six survey waves, and two rounds of SARS-CoV-2 antibody testing with fingerpricks. It was also reviewed and approved by our Office of Human Subjects.

We have updated the text to clarify that $30 was the total possible compensation for all the study procedures from the longitudinal parent study. We now note that the compensation for the study procedures giving rise to the baseline data used in this study was $10. We hope this alleviates concerns that we might have been overcompensating participants for a single survey.

New text found under the Materials and methods -study design section: “Participants received up to $30 for completing all the parent study activities, from which they would have received $10 for completing the baseline survey information used in this study.”

Page 6, lines 121-123

12. "Keywords should be written in alphabetical order. The use of mental health as keyword do not add any information as the mental health symptoms dealt with were only depressive symptoms and perceived stress. 

Response: We appreciate the reviewer’s comment about excluding mental health from the list of keywords. As suggested, we removed the term “mental health” from the list and the remaining keywords are in alphabetical order. As our fourth key word, we added “university students” to account for the population our study focuses on. 

Page 2, line 48

13. Background lines 79-80, needs citation 

Response: Thank you to the reviewer for highlighting the need for a reference. To support the statement, “These reductions in substance use could be suggestive of fewer opportunities for social gatherings among students as a result of COVID-19 prevention policies,” we have now cited the following reference: 

Layman HM, Thorisdottir IE, Halldorsdottir T, Sigfusdottir ID, Allegrante JP, Kristjansson AL. Substance Use Among Youth During the COVID-19 Pandemic: a Systematic Review. Curr Psychiatry Rep. 2022;24(6):307-24.

doi: 10.1007/s11920-022-01338-z

Page 4, lines 78-80

14. Objectives lines 100-103, is not objective it should be moved to introduction section. In addition there are several studies on mental health problems among students all over the globe. 

We have revised and relocated the sentences (“COVID-19 mental health studies have primarily examined the relationships between behaviors and mental health symptoms during the beginning months of the pandemic [18, 20-25]. Few have assessed US college students’ mental health in Fall 2020 during the COVID-19 pandemic as it relates to social factors and COVID-19 protective behaviors [26, 27].”) originally found at the start of the objectives to the end of the background section (page 5, lines 100-104). 

As for the second portion of the comment, we have addressed it in an earlier mention of the comment above at comment #6. We have pasted it below for quick review:

Although there are many studies that have focused on mental health problems among students, few have focused on the relationship between the social factors and COVID-19 protective behaviors (staying at home from work/school and avoiding social events) and the mental health of university students. We have revised the sentences in the introduction and discussion to delineate the contribution of our study to existing literature.

Revised statement in the introduction: “There is a need for this research because the pandemic is still ongoing, and we continue to be at risk for future pandemics. Although there is substantial existing literature on mental health and college students, there is a research gap around sociodemographic and behavioral activities and their relationship with students’ mental health during the pandemic context. This knowledge is valuable to support university efforts focused on improving the mental health of college students.”

Page 5, lines 105-110

15. Lines 216 to 217 is incomplete sentence. 

Response: We have gone ahead and revised the sentence to improve readability. It now reads as, “In contrast, the number of people the students “hung out” with while drinking alcohol was negatively associated with depressive symptoms.”

Page 15, lines 235-236 

16. Line 265 “key results” is unnecessary sub heading, check journal guideline. 

Response: As per the reviewer’s preference, we removed the “key results” header we originally had in the discussion section.

17. Line 275 to 276 repeating the objective once again in the discussion section is meaningless. We agree with the reviewer and have now removed the text highlighting the objective that we had in the discussion section.

18. Several limitations have been mixed up with the strength of the study section in the discussion. 

Response: We appreciate the reviewer’s comment regarding the current organization of the limitations section. We have re-organized the text by moving up the limitations that previously followed the strengths mentioned: 

Another limitation of this study was the use of self-reported data collected via an online survey instrument. There is a possibility of bias due to under-reporting for some of our key selected variables. For instance, due to the legal age of tobacco and nicotine products having been raised to 21 matching the legal age of consumption for alcohol, there may have been some under-reporting of substance use of individuals between age 18 and 20 [40]. However, self-response surveys for nicotine use and mental health symptoms are conventional for this type of research [41, 42]. Further, an additional limitation of our cross-sectional design of the study is the inability to capture the fluidity of mental health symptoms. Lastly, unlike in the Charles et al. study [26], there were no pre-pandemic matched mental health scores to compare changes in symptoms among our sample. 

During the regular college experience, college students experience a breadth of challenges, whether personal, academic, financial, or otherwise. The COVID-19 pandemic worsened some of those existing challenges and introduced new ones. Having conducted this study at the start of the Fall 2020 academic semester, as students returned from an all-online curriculum to a hybrid learning mode, is not representative of all the different stages of the pandemic. However, it offers some insight into students’ mental health and well-being for future public health crises.

Pages 19-20, lines 337-353

19. Tables should be named appropriately i.e. play and year has to be mentioned. 

Response: We thank the reviewer’s feedback regarding the titles used for Tables 1 and 2. As per their feedback we have updated the title for Table 1 to include Fall 2020 and the title for Table 2 to mention university students from a midwestern US university and the time Fall 2020. 

Table 1

Original title: Table 1. Depressive Symptoms and Perceived Stress Scores in a Sample of Undergraduate Students from a Midwestern University

New title: Table 1. Depressive Symptoms and Perceived Stress Scores in a Sample of Undergraduate Students from a Midwestern US University, Fall 2020

Page 11

Table 2

Original title: Table 2. Results for Logistic Regression for Depressive Symptoms (CES-D-10) and Linear Regression for Stress Symptoms (PSS) by Predictors (95% CI)

New title: Table 2. Results for Logistic Regression for Depressive Symptoms (CES-D-10) and Linear Regression for Stress Symptoms (PSS) by Predictors (95% CI), Undergraduate Students from a Midwestern US University, Fall 2020 

Page 13

20. Reviewer #2: This paper is very interesting and easy to read. However, I recommend to improve the discussion and conclusion sections. The discussion should be in-depth and contains not only summaries of research results, but above all their significance and implications for university practice 

Response: We thank Reviewer #2 for the feedback and suggestions. We have addressed a similar comment above under Reviewer #1’s comment #9. For quick reference, we have copied and pasted the response below:

The findings of this current study may have some implications for future university public health communications and prevention efforts. Universities need to provide opportunities for social interaction that maintain safety from infectious disease transmission. For example, this may include hosting social activities in an outdoor space where students can maintain a safe distance from each other while interacting with one another, rather than staying in their dorm or apartment in social isolation. This may also include establishing peer support groups that regularly meet and provide each other with guidance on how to safely socialize. Our study contributes to the existing mental health and COVID-19 among college student literature by identifying a negative relationship between the recommended refrain of social activities and protective behaviors and two mental health outcomes: depression and stress during a time when the requirements were no longer as strongly enforced. 

Future research could further investigate the relationship between these social factors, COVID-19 protective behaviors, and the mental health of students who, now years into the pandemic, have continued to refrain from social activities. Furthermore, additional research could review university pandemic incident response plans and the available support systems and existing interventions aimed at preventing depression and stress.

Page 20-21, lines 360-377

Additional Journal Requirements 

and 

 Response: We have reviewed the PLOS ONE style requirements and file naming and have ensured that the manuscript and file names meet the requirements.

22. Please state the full name of the Institutional Review Board that approved your study. 

Response: We updated the Methods-study design section in the manuscript to state the full name of the Institutional Review Board to now include “Indiana University Human Subjects Office.”

23. You indicated that you had ethical approval for your study. In your Methods section, please ensure you have also stated whether you obtained consent from parents or guardians of the minors included in the study or whether the research ethics committee or IRB specifically waived the need for their consent. 

Response: There were no minors included in this study. To be eligible, all participants had to meet the minimum age requirement of 18. Therefore, since minors were not eligible, we did not include any text on obtaining consent from parents or guardians for minors.

24. In your Data Availability statement, you have not specified where the minimal data set underlying the results described in your manuscript can be found. PLOS defines a study's minimal data set as the underlying data used to reach the conclusions drawn in the manuscript and any additional data required to replicate the reported study findings in their entirety. All PLOS journals require that the minimal data set be made fully available. For more information about our data policy, please see http://journals.plos.org/plosone/s/data-availability.

 Response: We provide the minimal data set underlying the results described in the manuscript as a supplemental document (Excel file).

25. Your ethics statement should only appear in the Methods section of your manuscript. If your ethics statement is written in any section besides the Methods, please delete it from any other section. 

Response: We removed the ethics statement from the of the manuscript and now it is found only in the Methods section.

---

## [Decision Letter · Decision Letter 1]

6 Dec 2022

The association between social factors and COVID-19 protective behaviors and depression and stress among midwestern US college students

PONE-D-22-20107R1

Dear Dr. Garcia Colato,

We’re pleased to inform you that your manuscript has been judged scientifically suitable for publication and will be formally accepted for publication once it meets all outstanding technical requirements.

Kind regards,

Michio Murakami

Academic Editor

PLOS ONE

Additional Editor Comments (optional):

Reviewers' comments:

Reviewer's Responses to Questions

**Comments to the Author**

1. If the authors have adequately addressed your comments raised in a previous round of review and you feel that this manuscript is now acceptable for publication, you may indicate that here to bypass the “Comments to the Author” section, enter your conflict of interest statement in the “Confidential to Editor” section, and submit your "Accept" recommendation.

Reviewer #1: All comments have been addressed

2. Is the manuscript technically sound, and do the data support the conclusions?

Reviewer #1: Yes

3. Has the statistical analysis been performed appropriately and rigorously? 

Reviewer #1: Yes

4. Have the authors made all data underlying the findings in their manuscript fully available?

Reviewer #1: Yes

5. Is the manuscript presented in an intelligible fashion and written in standard English?

Reviewer #1: Yes

6. Review Comments to the Author

Reviewer #1: (No Response)

7. PLOS authors have the option to publish the peer review history of their article (what does this mean?). If published, this will include your full peer review and any attached files.

Reviewer #1: **Yes: **Henok Dagne Derso

---

## [Editor Report · Acceptance letter]

12 Dec 2022

PONE-D-22-20107R1 

The association between social factors and COVID-19 protective behaviors and depression and stress among midwestern US college students 

Dear Dr. Garcia Colato:

I'm pleased to inform you that your manuscript has been deemed suitable for publication in PLOS ONE. Congratulations! Your manuscript is now with our production department. 

Kind regards, 

on behalf of

Dr. Michio Murakami 

Academic Editor

PLOS ONE